# Prediction of Mechanical Behaviours of FRP-Confined Circular Concrete Columns Using Artificial Neural Network and Support Vector Regression: Modelling and Performance Evaluation

**DOI:** 10.3390/ma15144971

**Published:** 2022-07-17

**Authors:** Pang Chen, Hui Wang, Shaojun Cao, Xueyuan Lv

**Affiliations:** 1School of Civil and Transportation Engineering, Hebei University of Technology, Tianjin 300401, China; 2020029@hebut.edu.cn (P.C.); 202031603007@stu.hebut.edu.cn (H.W.); 2China Construction First Group Construction & Development Co., Ltd., Beijing 100102, China; lvxueyuan@chinaonebuild.com

**Keywords:** FRP-confined circular concrete columns, artificial neural network, support vector regression, mechanical behaviours

## Abstract

The prediction and control of the mechanical behaviours of fibre-reinforced polymer (FRP)-confined circular concrete columns subjected to axial loading are directly related to the safety of the structures. One challenge in building a mechanical model is understanding the complex relationship between the main parameters affecting the phenomenon. Artificial intelligence (AI) algorithms can overcome this challenge. In this study, 298 test data points were considered for FRP-confined circular concrete columns. Six parameters, such as the diameter-to-fibre thickness ratio (*D/t*) and the tensile strength of the FRP (*f*_frp_) were set as the input sets. The existing models were compared with the test data. In addition, artificial neural networks (ANNs) and support vector regression (SVR) were used to predict the mechanical behaviour of FRP-confined circular concrete columns. The study showed that the predictive accuracy of the compressive strength in the existing models was higher than the peak compressive strain for the high dispersion of material deformation. The predictive accuracy of the ANN and SVR was higher than that of the existing models. The ANN and SVR can predict the compressive strength and peak compressive strain of FRP-confined circular concrete columns and can be used to predict the mechanical behaviour of FRP-confined circular concrete columns.

## 1. Introduction

Fibre-reinforced polymers (FRPs) are widely used in composite structures and for structural strengthening owing to their light weight, high strength, and good corrosion resistance [1,2,3,4,5,6,7]. As shown in Figure 1, an FRP-confined circular concrete column consists of core concrete and an external confinement of wrapped FRP. $\sigma_{c}$ is the axial compressive stress on the FRP-confined circular concrete column; $\sigma_{f}$ is the lateral constraint stress on the core concrete of the FRP-confined circular concrete column; and $\sigma_{\mathrm{cf}}$ is the reaction force generated by the core concrete under the FRP-confined action. 

In FRP-confined circular concrete columns, the core concrete is subjected to triaxial compression under axial loading, which improves the bearing capacity and ductility of the members and subsequently reduces the section size and self-weight of the members [8,9,10,11]. In addition, external confinement by the wrapping of FRP protects the core concrete from corrosion and improves the durability of the members. Relevant research has shown that the restraining action of FRP can increase the bearing capacity and ductility of core concrete by 2–46% and 14–923%, respectively [12,13,14,15]. Therefore, studying the axial compressive mechanical behaviours of FRP-confined circular concrete columns is vital in expanding their application in engineering practices. 

Researchers have extensively investigated the axial compressive mechanical behaviours of FRP-confined concrete columns. They have studied the influences of the diameter-to-fibre thickness ratio (*D/t*), the compressive strength of the core concrete (*f*_co_), the strain corresponding to the compressive strength of core concrete (*ε*_co_), the tensile strength of FRP (*f*_frp_), the elastic modulus of FRP (*E*_frp_), and the ultimate tensile strain of FRP (*ε*_frp_) on the axial compressive mechanical behaviours of FRP-confined circular concrete columns and proposed their axial compression constitutive models. Fardis studied the influences of the section size and compressive strength of core concrete on the axial compressive mechanical behaviours of FRP-confined circular concrete columns and proposed a corresponding axial compression constitutive model based on the test results [16]. Saadatmanesh established an axial compression constitutive model of CFRP- and GFRP-confined circular concrete columns based on the steel-confined concrete model proposed by Mander [17,18]. Nanni studied the influence of the FRP type on the axial compressive mechanical behaviours of FRP-confined circular concrete columns and proposed their hyperbolic constitutive model based on the test results [19]. Samaan studied the influence of FRP layers on the axial compressive mechanical behaviours of GFRP-confined circular concrete columns and proposed an axial compression constitutive model [20]. Lam and Teng proposed a two-stage constitutive model of FRP-confined circular concrete columns based on experimental data, which combined a parabola and a straight line [21,22,23,24]. Chastre and Silva studied the influences of the diameter-to-fibre thickness ratio, the elastic modulus of FRP, and the strength of core concrete on the axial compressive mechanical behaviours of GFRP-confined circular concrete columns and proposed an axial compression constitutive model of FRP-confined circular concrete columns based on the test results [25]. In order to reduce the error increase in the FRP constraint model due to FRP failure, Gian proposed an effective strain analysis model of FRP-confined cylinder concrete columns based on a theoretical analysis and the test data [26]. Wu and Wei studied the effects of the ultimate tensile strain of FRP and the strength of core concrete on the axial compressive mechanical behaviours of FRP-confined circular concrete columns and proposed an axial compression constitutive model of FRP-confined circular concrete columns based on the test results [27]. With the depth of research on FRP-confined concrete columns increasing, researchers have carried out a series of research works on FRP-confined non-circular concrete columns. Gian established a simplified model of FRP-confined circular concrete columns based on the iterative constraint model proposed by himself [28]. Jiang studied the effect of FRP restraint on reducing the strength degradation of square concrete columns based on a test under the eccentric loading of FRP-confined square concrete columns [29]. He reviewed the research status of FRP-confined non-circular concrete columns with modified shapes and proved the advantage of the curving section through the influence of key parameters on the FRP constraints [30]. All the existing models are semi-empirical and semi-theoretical formulations based on limited data, which do not consider the influence of all the parameters on the compressive strength and ultimate compressive strain of the FRP-confined circular concrete columns. The accuracy of the prediction needs to be evaluated; therefore, it is urgent to propose a reliable model that can accurately predict the axial compressive constitutive relationship of FRP-confined circular concrete columns.

In recent years, machine learning has shown unique advantages in solving various problems in structural engineering. A variety of machine learning methods have been widely applied in performance prediction, data classification, image recognition, structural simulation, etc. In particular, artificial neural networks (ANNs) and support vector regression have attracted the most attention [31,32,33,34,35,36,37,38].

ANNs are mathematical models that simulate the neural frame of a human to process complex information. They are widely used for data prediction [39,40]. Liu used an ANN and a swarm intelligence algorithm to predict the carbonation depth of recycled concrete through nine parameters, including temperature, recycled aggregate replacement rate, water absorption, and exposure time, and the results showed that the ANN performed better than the conventional formula [41]. Amiri used an ANN to predict the mechanical behaviours and durability of concrete containing constructional waste through the water-binder ratio, the recycled aggregate replacement rate, and other parameters. The results proved that the ANN had excellent accuracy in predicting the target value [42]. Jayasinghe established a shear test database of concrete beams without shear reinforcement and evaluated the forecast accuracy of the ANN and existing models. The results showed that the ANN was more accurate and is an effective tool for the influence analysis of a single parameter [43].

Support vector regression (SVR) is a regression model developed by Vapnik [44], who introduced insensitive loss functions into the support vector machine (SVM) based on statistical learning theory. It has better predictive ability. Ahmad used SVR, NMR (nuclear magnetic resonance), and an ANN to predict the splicing strength of reinforced concrete members based on the diameter of the rebar, the compressive strength of concrete, and the protective cover thickness. The results show that SVR has the highest forecasting accuracy [31]. Tran used SVR to predict the adhesion strength of the interface between FRP and concrete using the water–cement ratio, the recycled aggregate replacement rate, the sand rate, and other parameters. The results showed that SVR had good predictive performance [45].

ANNs and SVR have been successfully applied in the field of architecture and achieved ideal results [46,47,48,49,50]. Whether they can accurately predict the axial compressive mechanical behaviour of FRP-confined circular concrete columns needs to be evaluated. Based on this, our study collected the test data of the axial compressive mechanical behaviours of FRP-confined circular concrete columns from the relevant published literature and established a reliable database containing 298 datasets. *D/t*, *f*_co_, *ε*_co_, *f*_frp_, *E*_frp_, and *ε*_frp_ were set as input sets. Based on the established database, the forecast accuracy of the existing axial compression constitutive model for FRP-confined circular concrete columns was evaluated and analysed. ANN and SVR models were developed, and the forecast accuracy of the axial compressive mechanical behaviours of the FRP-confined circular concrete columns was evaluated and analysed using the ANN, SVR, and the existing axial compression constitutive models. Finally, based on the intelligent algorithm model, the affecting key factors were analysed using expanded parameters. Finally, the key factors affecting the axial compressive mechanical behaviours of the FRP-confined circular concrete columns were analysed based on an intelligent algorithm.

## 2. Database

The database relied on 298 specimens of tested data obtained from 13 studies [1,8,43,44,45,51,52,53,54,55,56,57,58]. The data information has been listed in Appendix A. It is crucial to collect appropriate parameters to study the axial compressive mechanical behaviour of FRP-confined circular concrete columns. Thus, such a database considers the influence of six parameters on the mechanical behaviour of FRP-confined circular concrete columns and the influence of *D/t*, *f*_co_, *ε*_co_, *f*_frp_, *E*_frp_, and *ε*_frp_ on the axial mechanical behaviour of FRP-confined circular concrete columns. Detailed information on the main parameters is presented in Table 1.

The range of the *D/t*, *f*_co_, *ε*_co_, *f*_frp_, *E*_frp_, and *ε*_frp_ are shown in Figure 2 and Figure 3. The *D/t* was mainly distributed in the range of 0–500. The minimum *f*_co_ was 9.9 MPa, the highest was 136.3 MPa, and the data between 50 MPa and 100 MPa were the highest. The *ε*_co_ was concentrated in the range of 1500–3000 με. The *f*_frp_ in the range of 2000 MPa–4000 MPa was up to 65%. Most data on the *E*_frp_ were distributed in the range of 200–300 GPa. The data points of the *ε*_frp_ in the range of 4.5 × 10^4^–5 × 10^4^ με were the least, and maximum data points were in the range of 3 × 10^4^–4.5 × 10^4^ με.

## 3. Predictive Models of the Axial Compressive Mechanical Behaviours of FRP-Confined Circular Concrete Columns

### 3.1. Evaluation Indices

To evaluate the predictive accuracy of the models of the axial compressive mechanical behaviours of the FRP-confined circular concrete columns, the regression coefficient (*R*^2^), the mean square error (*MSE*), the mean absolute percentage error (*MAPE*), and the integral absolute error (*IAE*) were used to evaluate the predictive accuracy of the models. *R*^2^ reflects the correlation between the independent and dependent variables. The closer *R*^2^ is to 1, the higher is the correlation between the predictive value and the actual value. The *MSE* reflects the average error. The lower the *MSE*, the smaller the error between the predictive value and the actual value. *MAPE* reflects the degree of data dispersion. The smaller the *MAPE*, the more the predictive value converges to the actual value. Additionally, the *IAE* reflects the predictive accuracy. The smaller the *IAE*, the higher the predictive accuracy of the data. All the indices are often used to evaluate the predictive accuracy of the neural network model [42,59,60,61,62]. The design formulae for each index are given in Equations (1)–(4).
(1)$$R^{2}=\frac{(\sum_{i=1}^{n}(O_{i}-{\bar{O}}_{i})(C_{i}-{\bar{C}}_{i}){)}^{2}}{\sum_{i=1}^{n}{\left(O_{i}-{\bar{O}}_{i}\right)}^{2}\sum_{i=1}^{n}{\left(C_{i}-{\bar{C}}_{i}\right)}^{2}}$$
(2)$$MSE=\frac{1}{n}\overset{n}{\underset{i=1}{\sum }}{\left(C_{i}-O_{i}\right)}^{2}$$
(3)$$MAPE=\frac{1}{n}\overset{n}{\underset{i=1}{\sum }}\left|\frac{C_{i}-O_{i}}{C_{i}}\right|$$
(4)$$IAE=\frac{\sqrt{\sum_{i=1}^{n}{\left(O_{i}-C_{i}\right)}^{2}}}{\sum_{i=1}^{n}C_{i}}\times 100\%$$
where $O_{i}$ is the actual value of the compressive strength or ultimate compressive strain (MPa/με); *C_i_* is the predictive value of the compressive strength or ultimate compressive strain (MPa/με); ${\bar{O}}_{i}$ is the average value of the actual value of the compressive strength or ultimate compressive strain (MPa/με); ${\bar{C}}_{i}$ is the average value of the predictive value of the compressive strength or ultimate compressive strain; and *n* is the number of data points.

### 3.2. Evaluation of Existing Axial Compression Constitutive Models

#### 3.2.1. Existing Axial Compression Constitutive Models 

At present, there is much research on the axial compressive mechanical behaviour of FRP-confined circular concrete columns. The influence of key parameters on the axial compressive mechanical behaviours of FRP-confined circular concrete columns has been studied, and many axial compression constitutive models of FRP-confined circular concrete columns have been proposed. Among them, the Mander [16], Fardis [17], Lam [21], Bisby [63], Wu [64], and Youssef [65] models are widely used.

The Mander model was proposed as a steel-confined concrete model in 1988 [16] and was later adopted by the guidelines for the selection, design, and installation of FRP systems for externally strengthening concrete structures to calculate the axial compressive mechanical behaviours of FRP-confined circular concrete columns [66]. Fardis experimentally studied the axial compressive mechanical behaviours of 46 FRP-confined circular concrete columns and proposed axial compression constitutive models of CFRP-confined circular concrete columns [17]. Lam and Teng proposed a constitutive model for FRP-confined circular concrete columns combined with a parabola with a straight line [21]. Bisby studied the influence of the FRP type on the axial compressive mechanical behaviours of FRP-confined circular concrete columns and proposed a constitutive model applicable to medium and weak FRP-confined concrete columns [63]. Wu Gang proposed a simplified trilinear constitutive model for FRP-confined circular concrete columns [64]. Youssef proposed an axial compressive constitutive model of FRP-confined circular concrete columns considering the influence of the FRP type and the diameter-to-fibre thickness ratio [65]. Detailed information on each model is shown in Table 2.

#### 3.2.2. Predictive Results of Existing Axial Compression Constitutive Models 

Based on the established database, the predictive accuracies of the six axial compressive constitutive models of FRP-confined circular concrete columns were compared and analysed. The results of the compressive strength of each model are shown in Figure 4.

As shown in the figure, most of the data points of the Mander [16] and Fardis [17] models are above a straight line, indicating that both the models overestimate the compressive strength of FRP-confined circular concrete columns. However, most of the data points of the Bisby [63] and Youssef [65] models are below the best fit line, indicating that both these models underestimate the compressive strength of FRP-confined circular concrete columns. The predictive capability of the Lam [21] and Wu [64] models is good, and the data points are uniformly distributed on both sides of the best fit line. Among them, the Lam model [21] has a more concentrated distribution of data points, and the Lam model [21] has the highest predictive accuracy of compressive strength for FRP-confined circular concrete columns.

The predictive results of the models for strain corresponding to the compressive strength of the FRP-confined circular concrete columns are shown in Figure 5. It can be observed from the figure that most of the data points of the Mander [16], Lam [21], and Wu [64] models are above the best fit line, indicating that these models overestimate the peak compressive strain of FRP-confined circular concrete columns. Most of the data points of the Fardis [17], Bisby [63], and Youssef [65] models are below the best fit line. This indicates that these models underestimated the peak compressive strain of FRP-confined circular concrete columns. The distribution of the data points in each figure is scattered, indicating that the predictive accuracy of the peak compressive strain in the existing axial compressive constitutive models of FRP-confined circular concrete columns is poor.

The evaluation indexes (*R*^2^, *MSE*, *MAPE*, and *IAE*) for the predictive accuracy of the compressive strength of the above models are listed in Table 3. It can be seen from Table 3 that the predictive accuracy of all the models is good, and the *R*^2^ is above 0.5. The Lam model [21] had the highest predictive accuracy, with an *R*^2^ of up to 0.83 and the lowest average error and a small dispersion.

Table 4 summarises the evaluation indices (*R*^2^, *MSE*, *MAPE*, and *IAE*) of the predictive accuracy in the peak compressive strain of all the models. It can be observed from Table 4 that the predictive accuracy of the peak compressive strain is lower than that of the compressive strength for all the models. The regression coefficient of the Fardis model [17] with the highest predictive accuracy was only 0.48; it had a high dispersion and error.

### 3.3. Evaluation of Predictive Models Based on ANN and SVR 

The predictive accuracy of the compressive strength and peak compressive strain by the existing axial compressive constitutive model of FRP-confined circular concrete columns is low and that of the peak compressive strain is especially poor. Therefore, a new predictive model is urgently required to predict the axial compressive mechanical behaviour of FRP-confined circular concrete columns.

#### 3.3.1. Machine Learning Models

##### Artificial Neural Networks (ANNs)

An ANN is an information processing system that simulates the structural and functional characteristics of a biological neural system [61]. It includes input, hidden, and output layers. Each of these layers contains some nodes, which are interconnected to the elements in the subsequent layers [67]. The accuracy and the precision of the ANN is highly dependent on the structure of the developed models as well as the model parameters. These parameters contain the number of nodes in the hidden layers, the momentum rate, the learning cycle, and the learning accuracy. The basic idea is that in each hidden layer node, the weighted inputs from the previous layer are added together and the deviation is added to the system, where the weight depends on the momentum rate, and the deviation depends on the learning accuracy. This combination is then passed through a nonlinear activation function to form the output of each hidden neuron [68]. The back-propagation algorithm is often used to train ANN. Training is defined as the procedure for finding the optimal weights of the network so that the prediction error is minimized. In the learning cycle, the results are back-propagated, and the weights and the bias are adjusted in such a way that the obtained error is minimized [69]. 

A MATLAB-based program with a graphical user interface (GUI) was developed to train and evaluate the ANN model. The ANN model divided the established database into two parts: 80% for training (general is 70–90%) and 20% for testing (general is 10–30%). According to the predictive target, there are six and one nodes in the input and output layers, respectively; the momentum rate is 0.5; the learning cycle is 10^3^; and the learning accuracy is 4 × 10^−^^7^.

The number of nodes in the hidden layer plays a vital role in the predictive accuracy of an ANN. To determine this, the predictive accuracies of different numbers of nodes in the hidden layers were compared and analysed. The influence of the number of nodes in the hidden layers on the predictive accuracy of the compressive strength is shown in Figure 6, and the influence of the number of nodes in the hidden layers on the predictive accuracy of peak compressive strain is shown in Figure 7. It can be seen that when the number of nodes in the hidden layers is 10, the *MSE* of the compressive strength and peak compressive strain is the lowest, and *R*^2^ is the highest. The predictive accuracy of the ANN model was the highest. Therefore, when predicting the compressive strength and peak compressive strain, the number of nodes in the hidden layers in the ANN model was 10.

##### Support Vector Regression (SVR)

The SVM is an intelligent algorithm for general classification problems, first proposed by Boser in 1992 [70]. Unlike traditional neural networks based on empirical risk minimisation, SVR is based on the principle of structural risk minimisation, which aims to minimise the upper bound of the expected risk and avoids reliance on the designer’s empirical knowledge. Vapnik introduced an insensitive loss function into the SVM to form a support vector regression (SVR) [44].

For the SVR model, the established database was divided into two parts: 80% for training and 20% for testing. The SVR model involves three parameters: ε (insensitive loss function), C (regularised constant), and g (kernel coefficient). In SVR, the main goal is to obtain a function that differs at most ε from the actual targets for all training data, while being as flat as possible [71]. The smaller the ε, the smaller the error of the regression function and the higher the degree of model fitting, where C is the regularized constant specified by the user. It is defined as the penalty factor to indicate the trade-off between the flatness of the function and the empirical error. C was mainly used to prevent overfitting. The higher the C, the more the samples with a training error greater than ε are punished, and the stronger is the predictive ability [72]. g is the kernel coefficient. The choice of kernel function is closely related to the performance of the SVR. The commonly used kernel functions in the regression include the linear kernel function, the polynomial kernel function, the radial basis function (RBF), and the sigmoid kernel function. Considering the infinite dimensional feature space corresponding to the RBF, the RBF is adopted in this study. Its expression is shown in Equation (5).
(5)$$K\left(x_{i},x_{j}\right)=e^{-g{||x_{j},-,x_{j}||}^{2}}$$
where $x_{i},x_{j\;}$ is the input vector, and *g* is the key parameter of the RBF which can affect the smoothness of the function. The larger the *g*, the better the predictive effect of the training set.

To determine the optimal values of ε, C, and g in the SVR, each value is tested in a certain space on the premise of specifying the step. Then, based on the flow of the SVR algorithm, the parameter values of the SVR with optimal accuracy are derived: ε = 0.01, C = 20, and g = 0.3. The flow of the SVR algorithm is illustrated in Figure 8.

#### 3.3.2. Predictive Results and Discussion of ANN and SVR 

Based on the established database, the comparison between the ANN and SVR models for the predictive results of compressive strength and peak compressive strain are shown in Figure 9, Figure 10, Figure 11 and Figure 12, respectively. It can be seen from Figure 9, Figure 10, Figure 11 and Figure 12 that the ANN and SVR models can accurately predict the compressive strength and peak compressive strain of FRP-confined circular concrete columns, and the predictive accuracy is much higher than that of their existing axial compressive constitutive models.

Among the existing axial compressive constitutive models for FRP-confined circular concrete columns, the Lam [21] and Fardis [17] models have the highest predictive accuracies for the compressive strength and peak compressive strain of FRP-confined circular concrete columns, respectively. The evaluation indexes of the predictive accuracy for the ANN, SVR, Lam [21], and Fardis [17] models are listed in Table 5. It can be seen that the predictive accuracy of the SVR model for the compressive strength and peak compressive strain of FRP-confined circular concrete columns is slightly higher than that of the ANN model and far higher than that of the Lam [21] and Fardis [17] models. In the SVR model, the *R*^2^ of the compressive strength and peak compressive strain is up to 0.96 and 0.94, respectively. In conclusion, the SVR model proposed in this study can provide an approximate basis for revising and unifying the compressive strength and peak compressive strain formulas for FRP-confined circular concrete columns.

## 4. Parameter Analysis

An ANN was used to accurately analyse the influence of *D/t*, *f*_co_, *f*_frp_, *E*_frp_, *f*_cc_/ *f*_co_, and *ε*_cc_/*ε*_co_ on the axial compressive mechanical behaviour of FRP-confined circular concrete columns. While analysing the influence of each parameter, the values of the other parameters were set as the average value of each parameter in the established database.

The influence of *D/t* on the compressive strength and peak compressive strain of the FRP-confined circular concrete columns is shown in Figure 13. It can be seen that both the compressive strength and the peak compressive strain of FRP-confined circular concrete columns decrease with an increase in the *D/t* of the specimen. When the *D/t* increased from 200 to 1000, the compressive strength decreased from 130 MPa to 60 MPa and the peak compressive strain decreased from 3 × 10^4^ to 10^4^. When the *D/t* increases and the section size of the core concrete remains unchanged, the fibre thickness decreases and the force of constraint it can provide decreases, thus reducing the bearing capacity and ductility of the specimens [73,74,75,76,77].

The influence of the *E*_frp_ on the compressive strength and peak compressive strain of the FRP-confined circular concrete columns is shown in Figure 14. It can be seen that the compressive strength of the FRP-confined circular concrete columns is positively correlated with the *E*_frp_, whereas the peak compressive strain is negatively correlated with the *E*_frp_. When the *E*_frp_ increases from 40 GPa to 200 GPa, the compressive strength increases from 90 MPa to 150 MPa, and the peak compressive strain decreases from 2.5 × 10^4^ to 1.5 × 10^4^. When the *E*_frp_ increases, the elastic deformation resistance, the brittleness of the specimens, and the bearing capacity of the specimens increase, and the ductility decreases [12,78,79].

The influence of *f*_frp_ on the compressive strength and peak compressive strain of the FRP-confined circular concrete columns is shown in Figure 15. It can be observed that the compressive strength and the peak compressive strain of the FRP-confined circular concrete columns increase with an increase in *f*_frp_. When *f*_frp_ increases from 800 MPa to 4000 MPa, the compressive strength increases from 50 MPa to 250 MPa, and the peak compressive strain increases from 8 × 10^3^ to 4 × 10^4^. When *f*_frp_ increases, the force of the constraint provided by the FRP increases, and the bearing capacity and the ductility of the specimens are improved [51,80,81].

The influence of *f*_co_ on the compressive strength and peak compressive strain of the FRP-confined circular concrete columns is shown in Figure 16. It can be seen that the compressive strength of the FRP-confined circular concrete columns is positively correlated with *f*_co_, while the peak compressive strain of the FRP-confined circular concrete columns is negatively correlated with *f*_co_. When *f*_co_ increases from 40 MPa to 100 MPa, the compressive strength of the FRP-confined circular concrete columns increases from 90 MPa to 160 MPa, and their peak compressive strain drops from 1.6 × 10^4^ to 4 × 10^3^. When *f*_co_ increased, the overall strength, brittleness, and the bearing capacity of the specimens increased, and the ductility decreased [82,83,84].

The influence of the mechanical behaviours of the core concrete on the FRP-confined circular concrete columns is shown in Figure 17. It can be seen that *f*_cc_/*f*_co_ is negatively correlated with *f*_co_, while *ε*_cc_/*ε*_co_ is positively correlated with *ε*_co_. When *f*_co_ increases from 40 MPa to 100 MPa, *f*_cc_/*f*_co_ drops from 2.5 to 1.5. When *ε*_co_ increases from 3000 με to 6000 με, *ε*_cc_/*ε*_co_ increases from 4 to 5. When *f*_co_ increases, its ductility decreases, and the restraint of FRP on the core concrete weakens. Additionally, when *ε*_co_ increases, its ductility increases. The restraint effect of FRP on core concrete is strengthened.

## 5. Conclusions

This study evaluated the existing axial compressive constitutive model of FRP-confined circular concrete columns and established a predictive model for the axial compressive mechanical behaviours of FRP-confined circular concrete columns based on ANNs and SVR. The main conclusions are as follows.

(1)A database of the axial compressive mechanical behaviours of FRP-confined circular concrete columns was established from the relevant published literature, which included 298 datasets. The effects of *D/t*, *f*_co_, *ε*_co_, *f*_frp_, *E*_frp_, and *ε*_frp_ on the axial mechanical behaviours of FRP-confined circular concrete columns were considered.(2)Comparing and analysing the existing axial compressive constitutive model of FRP-confined circular concrete columns, the Lam model [21] exhibited the highest predictive accuracy for the compressive strength of FRP-confined circular concrete columns.(3)Comparing and analysing the existing axial compressive constitutive models of FRP-confined circular concrete columns, the Fardis model [17] exhibited the highest predictive accuracy for the peak compressive strain of FRP-confined circular concrete columns.(4)The ANN and SVR models can be used to predict the axial mechanical behaviours of FRP-confined circular concrete columns. Their predictive accuracy was much higher than that of the existing axial compressive constitutive model of FRP-confined circular concrete columns, and SVR had the highest predictive accuracy.(5)An intelligent algorithm was used to analyse the parameters of the axial mechanical behaviour of FRP-confined circular concrete columns. The analysis results show that *D/t* is positively correlated with the compressive strength, whereas *f*_co_, *f*_frp_, and *E*_frp_ are negatively correlated with the compressive strength of the FRP-confined circular concrete columns. The *f*_frp_ is positively correlated with the peak compressive strain, whereas *D/t*, *E*_frp_, and *f*_co_ are negatively correlated with the peak compressive strain of FRP-confined circular concrete columns.

## Figures and Tables

**Figure 1 materials-15-04971-f001:**
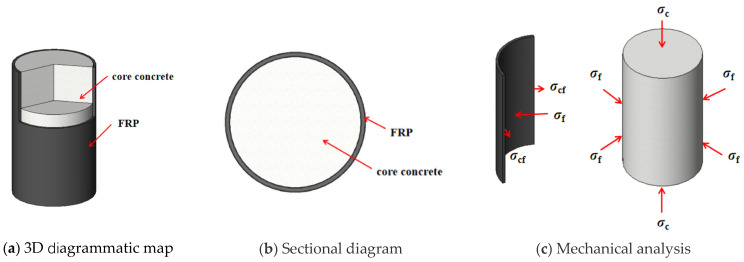
FRP-confined circular concrete column.

**Figure 2 materials-15-04971-f002:**
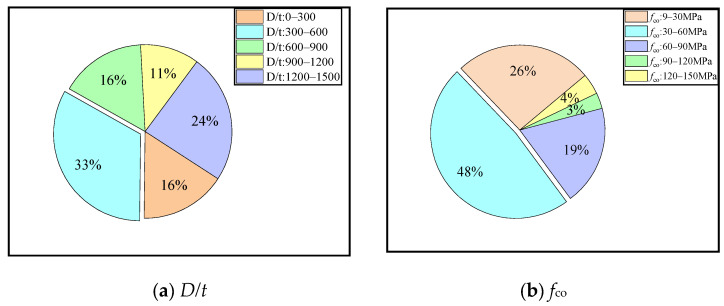
The value range of each parameter.

**Figure 3 materials-15-04971-f003:**
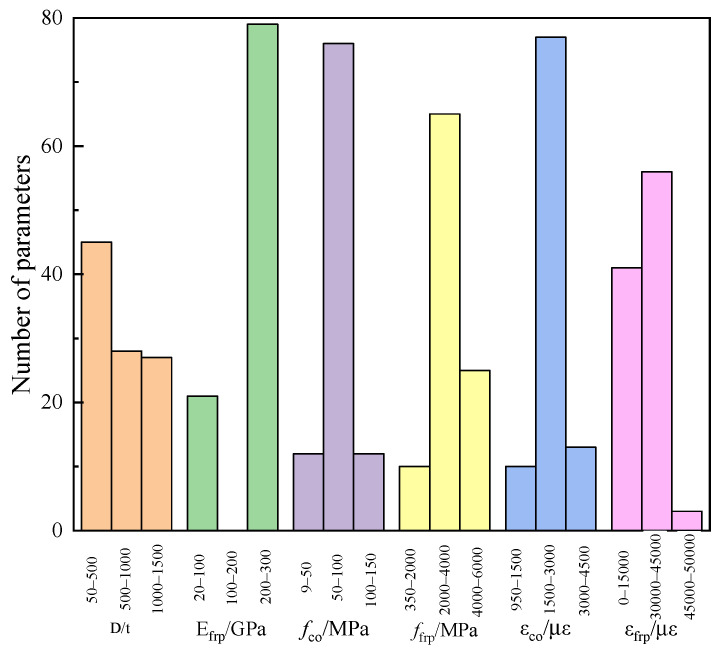
The histogram of each parameter distribution.

**Figure 4 materials-15-04971-f004:**
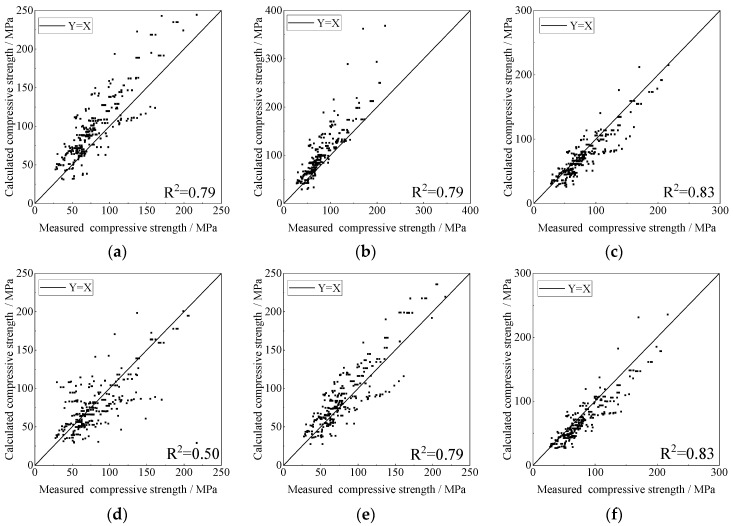
Predictive results of compressive strength. (**a**) Mander Model, (**b**) Fardis Model, (**c**) Lam Model, (**d**) Bisby Model, (**e**) Wu Model, (**f**) Youssef Model.

**Figure 5 materials-15-04971-f005:**
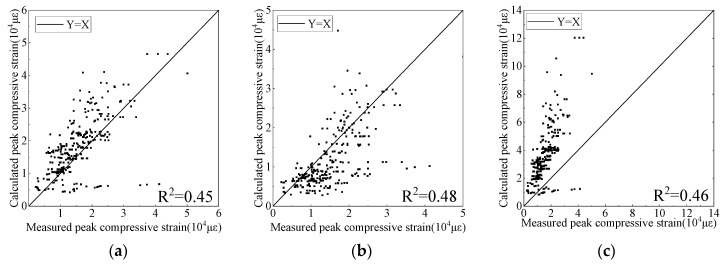
Predictive results of peak compressive strain. (**a**) Mander Model, (**b**) Fardis Model, (**c**) Lam Model, (**d**) Bisby Model, (**e**) Wu Model, (**f**) Youssef Model.

**Figure 6 materials-15-04971-f006:**
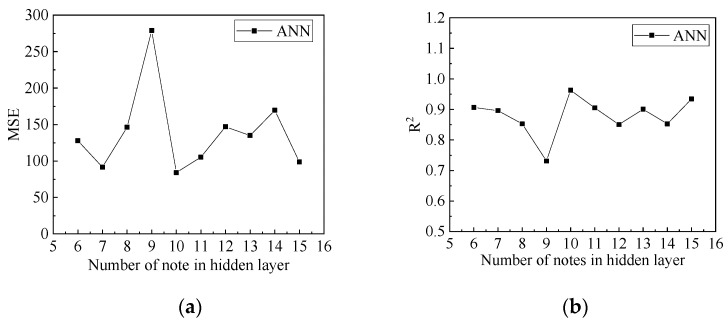
Influence of changes in the number of hidden layers on predictive accuracy of compressive strength: (**a**) the influence of changes in the number of hidden layers on *MSE*; (**b**) the influence of changes in the number of hidden layers on *R*^2^.

**Figure 7 materials-15-04971-f007:**
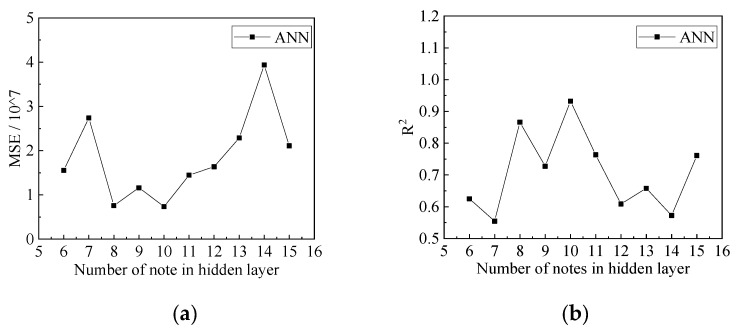
Influence of changes in the number of hidden layers on predictive accuracy of peak compressive strain: (**a**) The influence of changes in the number of hidden layers on *MSE*; (**b**) the influence of changes in the number of hidden layers on *R*^2^.

**Figure 8 materials-15-04971-f008:**
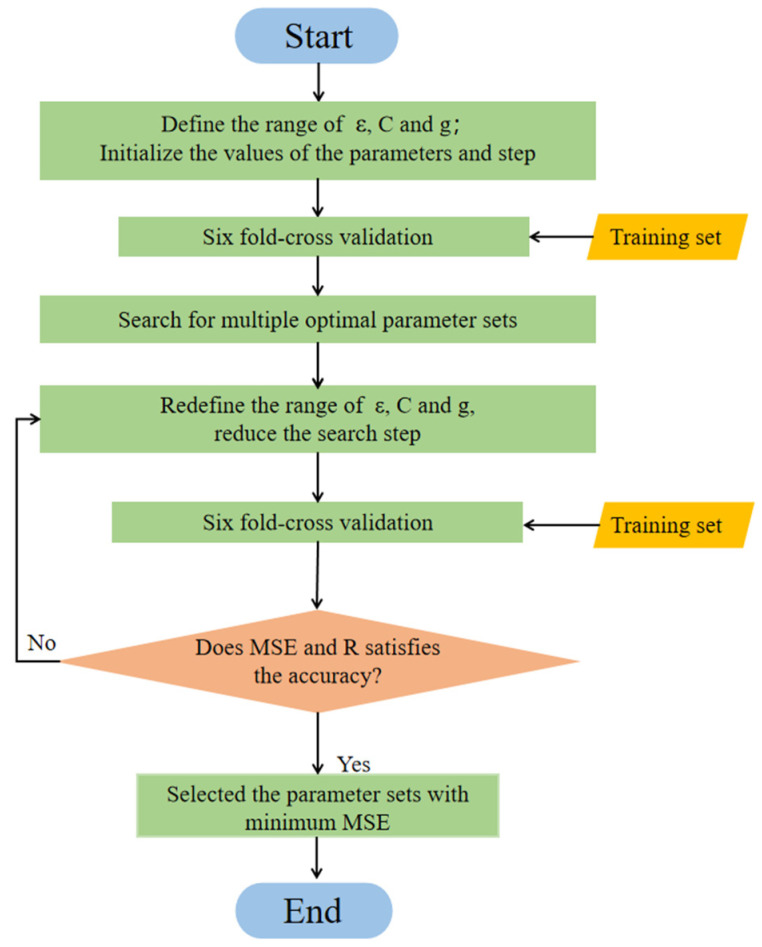
The SVR algorithm flow.

**Figure 9 materials-15-04971-f009:**
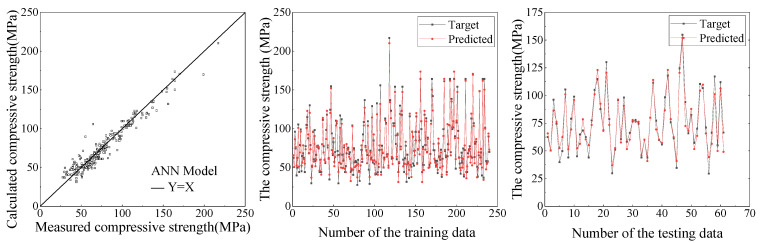
Predictive results of compressive strength by ANN.

**Figure 10 materials-15-04971-f010:**
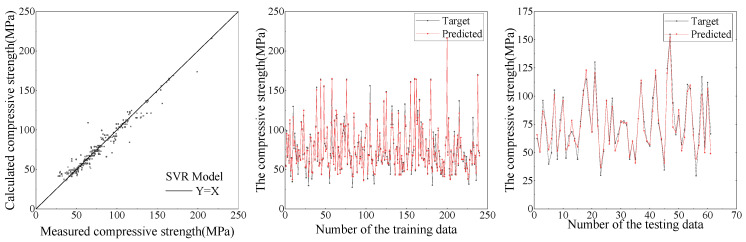
Predictive results of compressive strength by SVR.

**Figure 11 materials-15-04971-f011:**
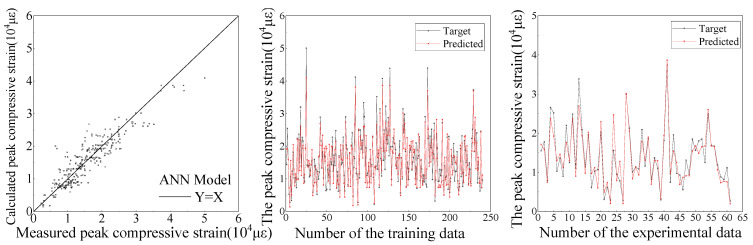
Predictive results of peak compressive strain by ANN.

**Figure 12 materials-15-04971-f012:**
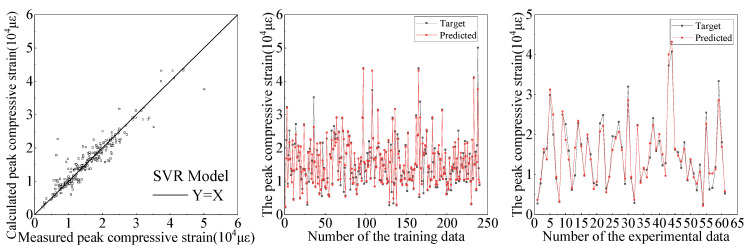
Predictive results of peak compressive strain by SVR.

**Figure 13 materials-15-04971-f013:**
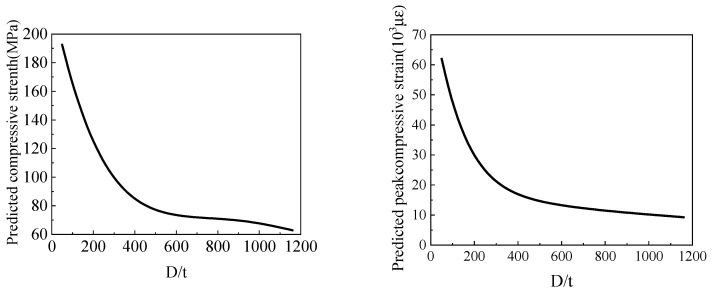
The influence of *D/t* on compressive strength and peak compressive strain.

**Figure 14 materials-15-04971-f014:**
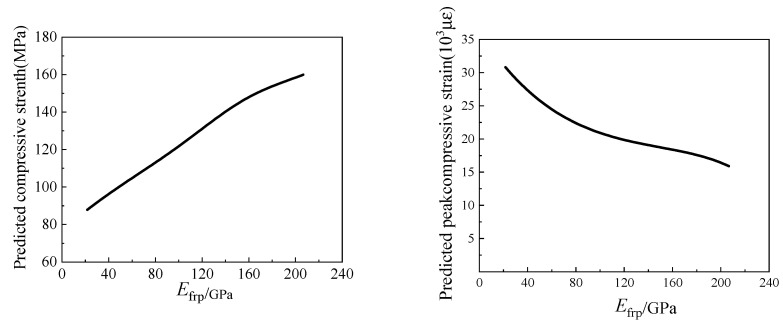
The influence of *E*_frp_ on compressive strength and peak compressive strain.

**Figure 15 materials-15-04971-f015:**
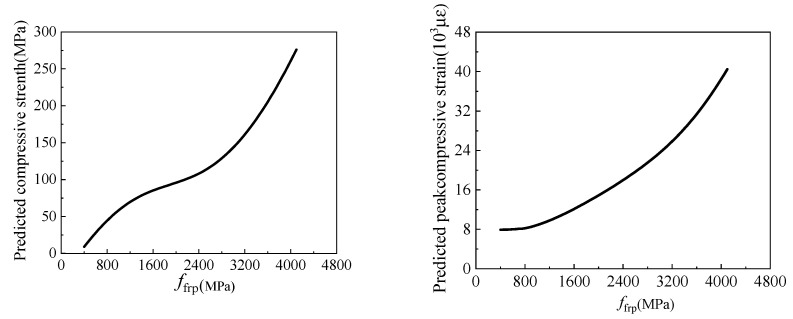
The influence of *f*_frp_ on compressive strength and peak compressive strain.

**Figure 16 materials-15-04971-f016:**
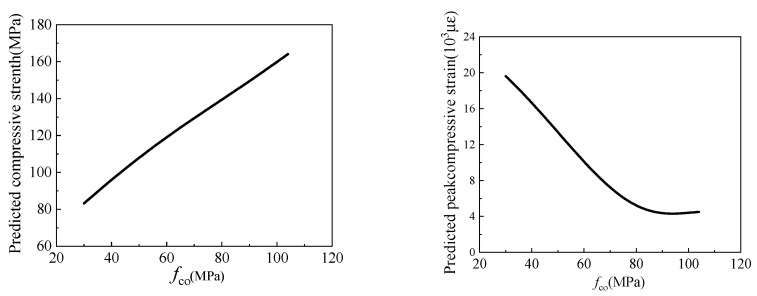
The influence of *f*_co_ on compressive strength and peak compressive strain.

**Figure 17 materials-15-04971-f017:**
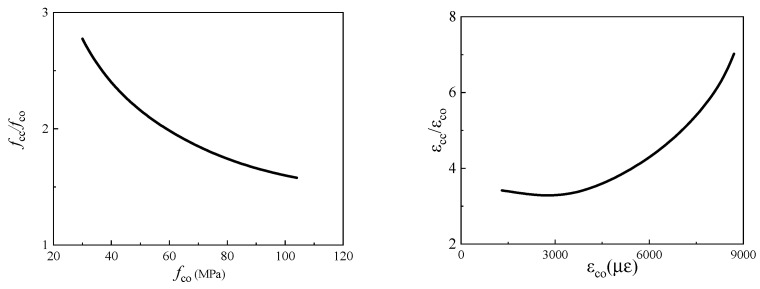
The influence of mechanical behaviours of core concrete on FRP-confined circular concrete columns.

**Table 1 materials-15-04971-t001:** The detailed information of main parameters.

Parameter	*D/t*	*f*_co_/MPa	*ε*_co_/με	*f*_frp_/MPa	*E*_frp_/GPa	*ε*_frp_/με
Minimum	57.3	9.9	950	383	21.6	7200
Median	615.4	39.4	2600	3500	234	16,000
Maximum	1500.0	136.3	3850	4933	245	43,000
Average	721.2	46.7	2440	3321	196.6	16,686
Standard deviation	433.1	25.6	598	1066	71.2	4967
Skewness	0.3	1.7	−0.9	−1.1	−1.3	3.9

**Table 2 materials-15-04971-t002:** Details of existing axial compression constitutive models.

Models	Equation for Stress	Equation for Strain	Parameters	Range of Application
Mander [16]	$\frac{f_{\mathrm{cc}}}{f_{\mathrm{co}}}=2.254\sqrt{1+\frac{7.94f_{l}}{f_{\mathrm{co}}}}-\frac{2f_{l}}{f_{\mathrm{co}}}-1.254$	$\frac{\varepsilon_{\mathrm{cc}}}{\varepsilon_{\mathrm{co}}}=5\frac{f_{\mathrm{cc}}}{f_{\mathrm{co}}}-4$	$f_{\mathrm{co}}/t_{\mathrm{frp}}/$styles of FRP	CFRP/GFRP-confined concrete columns
Fardis [17]	$\frac{f_{\mathrm{cc}}}{f_{\mathrm{co}}}=1+4.1\frac{f_{\mathrm{frp}}t_{\mathrm{frp}}}{Rf_{\mathrm{co}}}$	$\varepsilon_{\mathrm{cc}}=0.002+0.0005\frac{E_{\mathrm{frp}}t_{\mathrm{frp}}}{Rf_{\mathrm{co}}}$	R/styles of FRP	CFRP-confined circular concrete columns
Lam [21]	$\frac{f_{\mathrm{cc}}}{f_{\mathrm{co}}}=1+2\frac{f_{l}}{f_{\mathrm{co}}}$	$\frac{\varepsilon_{\mathrm{cc}}}{\varepsilon_{\mathrm{co}}}=15\frac{f_{l}}{f_{\mathrm{co}}}+2$	$f_{\mathrm{co}}/t_{\mathrm{frp}}/$ $\mathrm{styles}\;\mathrm{of}\;\mathrm{FRP}/f_{\mathrm{frp}}$	CFRP/GFRP-confined circular concrete columns
Bisby [63]	$\frac{f_{\mathrm{cc}}}{f_{\mathrm{co}}}=1+3.587{f_{r}}^{0.84}$	$\frac{\varepsilon_{\mathrm{cc}}}{\varepsilon_{\mathrm{co}}}=1+0.024\frac{f_{r}}{f_{\mathrm{co}}}$	styles of FRP	Medium and weak FRP-confined concrete column
Wu [64]	$\frac{f_{\mathrm{cc}}}{f_{\mathrm{co}}}=1.316+2.098\frac{f_{l}}{f_{\mathrm{co}}}-0317{\left(\frac{f_{l}}{f_{\mathrm{co}}}\right)}^{2}$	$\varepsilon_{\mathrm{cc}}=3.223\varepsilon_{\mathrm{frp}}{\left(\frac{f_{l}}{f_{\mathrm{co}}}\right)}^{0.44}$	$f_{\mathrm{co}}/t_{\mathrm{frp}}/$ styles of FRP/R	AFRP/CFRP/GFRP-confined circular concrete columns
Youssef [65]	$\frac{f_{\mathrm{cc}}}{f_{\mathrm{co}}}=1+2.25{f_{r}}^{1.25}$	$\varepsilon_{\mathrm{cc}}=0.003368+0.259\frac{f_{r}}{f_{\mathrm{co}}}{\left(\frac{f_{\mathrm{frp}}}{E_{\mathrm{frp}}}\right)}^{0.5}$	$t_{\mathrm{frp}}/$ styles of FRP/R	CFRP/GFRP-confined circular concrete columns

Note: $f_{\mathrm{cc}}$: compressive strength of FRP-confined circular concrete columns (MPa); $f_{\mathrm{co}}$: compressive strength of core concrete (MPa); $\varepsilon_{\mathrm{cc}}$: strain corresponding to compressive strength of FRP-confined circular concrete columns (με); $\varepsilon_{\mathrm{co}}$: strain corresponding to compressive strength of core concrete (με); $f_{\mathrm{frp}}$: tensile strength of FRP (MPa); $E_{\mathrm{frp}}$: elastic modulus of FRP (GPa); R: section radius of specimen (mm); $f_{r}$: effective lateral constraint stress provided by FRP (MPa); $f_{l}$: lateral constraint stress provided by FRP (MPa).

**Table 3 materials-15-04971-t003:** Predictive accuracy of compressive strength.

Models	*R* ^2^	*MSE*	*MAPE*	*IAE*/%
Mander [16]	0.79	788.5	0.22	1.64
Fardis [17]	0.79	1078.3	0.21	1.86
Lam [21]	0.83	271.9	0.20	1.30
Bisby [63]	0.50	707.5	0.27	1.99
Wu [64]	0.79	393.5	0.19	1.31
Youssef [65]	0.83	317.8	0.22	1.45

**Table 4 materials-15-04971-t004:** Predictive accuracy of peak compressive strain.

Models	*R* ^2^	*MSE*/10^7	*MAPE*	*IAE*/%
Mander [16]	0.45	5.45	0.39	2.36
Fardis [17]	0.48	5.69	0.61	3.48
Lam [21]	0.46	63.49	0.59	4.15
Bisby [63]	0.13	9.18	0.79	4.86
Wu [64]	0.15	47.10	0.52	3.97
Youssef [65]	0.45	5.49	0.33	2.81

**Table 5 materials-15-04971-t005:** Predictive accuracy analysis of compressive strength and peak compressive strain.

Performance Indices	Compressive Strength			Peak Compressive Strain		
	ANN	SVR	Lam [21]	ANN	SVR	Fardis [17]
*R* ^2^	0.92	0.96	0.83	0.87	0.94	0.48
*MAPE*	0.11	0.09	0.20	0.18	0.13	0.61
*IAE*/%	0.09	0.07	1.30	0.15	0.11	3.48
*MSE*	84.00	63.46	271.9	0.74 × 10^7^	0.47 × 10^7^	5.69 × 10^7^

## Data Availability

No new data were created or analyzed in this study. Data sharing is not applicable to this article.

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
