# Peer review of "Prediction of Mechanical Behaviours of FRP-Confined Circular Concrete Columns Using Artificial Neural Network and Support Vector Regression: Modelling and Performance Evaluation"

_materials, 2022, doi:10.3390/ma15144971_

Round 1
Reviewer 1 Report
Prediction of mechanical behaviours of FRP-confined circular concrete columns using artificial neural network and support vector regression: modelling and performance evaluation
The authors present ANN model and support vector regression model for predicting the compressive strength and the strain corresponding to the peak stress of FRP confined concrete as a better alternative to the mechanics based, or empirical or semi-empirical models proposed in the literature. The topic is timely and the content is going to be useful.
The main contributions of the authors are: 1. Creation of appropriate database, 2. Development of ANN model and 3. Development of a Support Vector Regression model.
This reviewer has the following queries which need to be addressed in revising the paper:
- Why the authors always use the word “column”. This makes the problem more complicated since the behaviour of column (confined or unconfined) depends on the slenderness ratio of the column. Has this been considered? Otherwise, what are the values for the test data considered needs to be mentioned.
- The strength and behaviour of FRP depends on the type, orientation and aspect ratio of fibres and the curing method adopted for producing FRP. These features for the test data considered needs to be documented.
- While creating the database it would have been better to classify the concretes as low, medium and high strength concrete and deal with them separately. Also, the depending on the aspect ratio of fibres classification could have been done as short and long fibres. Since no such classification has been done, it is possible that the other models considered in the paper might have not performed satisfactorily with the entire database. Thus, it is important to also indicate the range of applicability various constitutive relations considered in this study, as indicated by the authors in their original papers.
- There are minor technical problems with the figures that have been indicated in the MARKED COPY attached herewith review.
- The database can be made available as a supplementary file for this paper.
- The authors are encouraged to revise and resubmit the revised manuscript for re-review.

Reviewer 2 Report
This manuscript deals with prediction of the axial compressive mechanical behaviour of FRP-confined circular concrete columns based on ANN and SVR. The topic is of interest for the journal readers. It deserves publication however after some revisions according to next suggestions.
I suggest to improve the introduction with other references to previous works; the expansion from circular to non circular cross sections has been studied, and is a future development of this work (e.g. among others https://doi.org/10.3390/polym6041187 , https://doi.org/10.3390/polym14030564 ); the effect of D/t ratio, or more in general of premature failure of FRP or FRP efficiency (e.g among others https://doi.org/10.3390/polym12061261 , https://doi.org/10.1016/j.compositesb.2012.04.007 ). These are suggestions according to reviewer knowledge, authors feel free to consider others
I suggest to improve arrows direction in figure 1 for FRP.
At line 108, please support your statement “ANN and SVR have been successfully applied in the field of architecture and achieved ideal results” with one or more references.
I suggest to use different kind of diagrams for values of parameters (e.g. histograms, or bar charts).
I suggest to improve discussion of tables 3 and 4 by indicating the expected best performance for each index (e.g. R close to 1, lowest MSE, …)
Please provide at least a reference for momentum rate, learning cycle and learning accuracy definitions. The same for SVM and in particular: insensitive loss function, regularised constant, and kernel coefficient.
Please revise, in figure 8 there is MES or MSE?
Discussion in section 4 is not fully convincing to me; the impact of each parameter when fixing the others at the average value is informative, but it should be clarified how the impact of combined parameters work, so if possible, I suggest to use more than one case, so using also different combinations of the other parameters and not only one combination with average values.
For instance, figure 16 for predicted compressive strength is almost linear, as usually the predictive equations are based on normalized compressive strength (e.g. see table 2), however the core compressive strength is also inside the equations, so it is expected that confinement configuration, the fcc/fco is more than 2 for fco=40 MPa and decreases to about 1.6 for fco=100 MPa… Comments like this could improve the readability of the section 4.
Please provide more details of the “intelligent algorithm”.
Reviewer 3 Report
The present work shows the results of the Artificial Neural Networks and Support Vector Regression modelling of the mechanical characteristics of Fibre Reinforced Polymer concrete column from previously published papers. The study brings novelty to the field, but needs further clarification as follows.
- - Please, define the abbreviation FRP in the abstract and introduction sections.
- - Please make clear in which way is the here proposed model compared to the previous ones. In which format were the previous models you discuss given? Why are these new ANN and SVR better, except for r2? Did they use a mathematical software or not?
- Please, explain the claims in the abstract more deeply. (“The study showed that the predictive accuracy of the compressive strength in the existing models was higher than the peak compressive strain.”)
- The marks for the forces in Fig. 1 must be explained.
- Please, consider avoiding mentioning all the previous papers that used ANN for instance in concrete studies, and thus accumulation of references, like in [31-37], or similar. Refer to a few works that are crucial also for further discussion. Besides, improve the references list as stated in the journals` authors instruction and remove middle-sized brackets.
- The principles of ANN and SVR should be explained in a Methodology section and moved from the Introduction. After that, the goodness of fit tests like r2, MSE, MAPE and IAE should be theoretically explained (lines 151-169.). Also, the text from Chapters 3.3.1 and 3.3.2 should be merged and presented before any of the results.
- Define the abbreviation “NMR” (line 101).
- Please, delete the phrase in line 108 (“As the best authors know,”).
- Both abstract and the end of the Introduction section should show which were the parameters considered as inputs for the modelling. In which exactly is the difference between yours and previously obtained AI models? If the previous researchers used other parameters, those models cannot be compared by using coefficient of determination. Additionally, the previous studies did not predict this kind of concrete, so no comparing is possible.
- Which program is used for the modelling? Are the R values in Table 3 actually r2? I see no need for presenting the results from other studies, as they show other kinds of materials.
- Please, delete Fig. 5 as it is unnecessary and published in similar forms in all the books on ANN modelling.
- Please, read the instructions for authors and prepare the manuscript to follow the logical flow.
Round 2
Reviewer 3 Report
The manuscript has been significantlly improved and ready to be published.